# Uncertainty in tuberculosis clinical decision-making: An umbrella review with systematic methods and thematic analysis

**Francesca Wanda Basile** [1,2] *, **Sedona Sweeney** [2], **Maninder Pal Singh** [2], **Else Margreet Bijker** [1,3], **Ted Cohen** [4], **Nicolas A. Menzies** [5,6], **Anna Vassall** [2], **Pitchaya Indravudh** [2]

**1** Oxford Vaccine Group, Department of Paediatrics, University of Oxford, Oxford, United Kingdom, **2** Department of Global Health and Development, London School of Hygiene & Tropical Medicine, London, United Kingdom, **3** Department of Paediatrics, Maastricht University Medical Centre, MosaKids Children's Hospital, Maastricht, the Netherlands, **4** Department of Epidemiology of Microbial Diseases, Yale School of Public Health, New Haven, Connecticut, United States of America, **5** Department of Global Health and Population, Harvard TH Chan School of Public Health, Boston, Massachusetts, United States of America, **6** Center for Health Decision Science, Harvard TH Chan School of Public Health, Boston, Massachusetts, United States of America

* francesca.basile@paediatrics.ox.ac.uk

**Data Availability Statement:** All data relevant to replicate the study have been provided in the article or uploaded as supplementary information.

## Abstract

Tuberculosis is a major infectious disease worldwide, but currently available diagnostics have suboptimal accuracy, particularly in patients unable to expectorate, and are often unavailable at the point-of-care in resource-limited settings. Test/treatment decision are, therefore, often made on clinical grounds. We hypothesized that contextual factors beyond disease probability may influence clinical decisions about when to test and when to treat for tuberculosis. This umbrella review aimed to identify such factors, and to develop a framework for uncertainty in tuberculosis clinical decision-making. Systematic reviews were searched in seven databases (MEDLINE, CINAHL Complete, Embase, Scopus, Cochrane, PROSPERO, Epistemonikos) using predetermined search criteria. Findings were classified as barriers and facilitators for testing or treatment decisions, and thematically analysed based on a multi-level model of uncertainty in health care. We included 27 reviews. Study designs and primary aims were heterogeneous, with seven meta-analyses and three qualitative evidence syntheses. Facilitators for decisions to test included providers' advanced professional qualification and confidence in tests results, availability of automated diagnostics with quick turnaround times. Common barriers for requesting a diagnostic test included: poor provider tuberculosis knowledge, fear of acquiring tuberculosis through respiratory sampling, scarcity of healthcare resources, and complexity of specimen collection. Facilitators for empiric treatment included patients' young age, severe sickness, and test inaccessibility. Main barriers to treatment included communication obstacles, providers' high confidence in negative test results (irrespective of negative predictive value). Multiple sources of uncertainty were identified at the patient, provider, diagnostic test, and healthcare system levels. Complex determinants of uncertainty influenced decision-making. This could result in delayed or missed diagnosis and treatment opportunities. It is important to understand the variability associated with patient-provider clinical encounters and

**Funding:** Research reported in this publication was supported by the National Institute Of Allergy And Infectious Diseases of the National Institutes of Health (U01AI152084 to FWB, SS, MPS, TC, NAM, AV, PI). The funders had no role in study design, data collection and analysis, decision to publish, or preparation of the manuscript.

**Competing interests:** The authors have declared that no competing interests exist.

healthcare settings, clinicians' attitudes, and experiences, as well as diagnostic test characteristics, to improve clinical practices, and allow an impactful introduction of novel diagnostics.

## Introduction

Tuberculosis (TB) is a major infectious cause of morbidity and mortality globally. In 2022, 7.5 million people were diagnosed with TB, and 1.3 million people died because of the disease [1]. Missed or delayed TB diagnosis and treatment and low quality of care remain critical obstacles to disease control and improving health outcomes [2, 3].

To minimize diagnostic and treatment delays, high quality TB services should include access to rapid, affordable and accurate tests, such as the molecular WHO-recommended rapid diagnostics (mWRD) [4]. However, mWRD are seldom available at the point-of-care in resource-limited settings. Despite massive efforts to coordinate the global roll-out of GeneXpert (Cepheid, USA), recent data still show that this test is unavailable in many peripheral settings and more generally the underutilization of modern TB diagnostic technologies [5, 6].

The underutilization of diagnostics may arise due to a variety of factors, including as a consequence of providers' know-do gap [7]. This may become particularly evident in situations where care is tailored around the patient's perceived needs (e.g., clinicians offering a more affordable but less accurate diagnostic test), and best practices are not implemented (e.g., clinicians choosing quick symptom relief with low-cost pharmaceuticals over diagnostic certainty) [7]. Moreover, in resource-limited settings, when a patient presents with signs and symptoms suggestive of TB, clinicians may decide to start treatment based solely on clinical grounds, regardless of test availability [8, 9].

To standardize decision-making, pre- and post-test disease probabilities have been used to determine the thresholds for testing and treatment decisions [10, 11]. The provider determines a pre-test probability of disease, which varies depending on clinical signs and symptoms as well as the provider's experience, knowledge, and health care setting. The provider then decides whether to move forward with testing or initiating treatment. Following testing, the provider determines the post-test probability of disease and decides whether to start or withhold TB therapy [11].

There have also been multiple attempts to develop scoring systems and clinical prediction models for TB screening and diagnosis [12–16]. Scoring systems can help to calculate the probability of TB disease in a reproducible way and might be particularly helpful in paediatric TB, where currently available diagnostic tests lack high sensitivity. Additionally, clinical algorithms might help determine when testing is helpful and when a negative test is insufficient to withhold treatment [17].

However, in reality, the decision to test or treat presumptive TB cases can be affected by contextual variables beyond accessibility to diagnostics, or a mere computation of disease probability [18]. Provider characteristics, including their ability to cope with complexity, risk, and uncertainty, contribute to process variability [19]. Uncertainty is an inevitable component of clinical practice and can occur throughout the decision-making process: when formulating clinical hypotheses, identifying a diagnosis, choosing a test and interpreting its result, and interpreting patient preferences [20]. Multilevel models of uncertainty emphasize the dynamic interplay between different sources and types of uncertainty at each level, and may be useful to classify the challenges of clinical decision-making [20].

Understanding uncertainty in the TB decision-making process and the reasons why a provider would initiate empiric treatment or would not utilize a microbiological test even when available, is important to develop diagnostic tools that improve TB diagnosis and care behaviours and practices, and to project the impact of the introduction of novel diagnostic aids [21]. This umbrella review of systematic reviews (SR) aimed to identify factors influencing providers' decisions to test for TB, and initiate TB treatment in adult and paediatric patients with presumptive TB in high-TB and TB/HIV burden countries [22].

## Methods

### Study design rationale and methodology

An initial scoping search was conducted on MEDLINE (via OVID) for terms related to "tuberculosis" and "decision-making", and identified several reviews relevant to our research question [23–25]. Since most records evaluated either qualitative or quantitative primary studies, and often reported complementary findings, we chose an umbrella review design to allow for the inclusion of these reviews with a broad scope of inquiry and to achieve a higher level of synthesis [26–28].

The study was conducted in accordance with the Preferred Reporting Items for Systematic Reviews and Meta-Analyses (PRISMA) statement [29]. The Joanna Briggs Institute (JBI) guidelines for umbrella reviews [28, 30], and the Cochrane guidance for overviews of reviews [31] were also followed to address the specific issues arising when conducting umbrella reviews. The methodology of this review was prespecified in a protocol [32].

### Search strategy

Using a combination of key terms to maximize sensitivity, seven electronic databases were searched: MEDLINE (via OVID), CINAHL Complete, Embase, Scopus, Cochrane Central, the PROSPERO register, and Epistemonikos database. The search was limited from January 2007 (considering that the development Xpert MTB/RIF was completed in 2009) to the date of search, which was the 4th of July 2022. The search was rerun on the 21st of July 2023. We developed a comprehensive list of keywords and synonyms for each broad domain: 1) TB, 2) clinical decision-making. Terms were searched individually first and then combined using Boolean operators. The search was piloted in MEDLINE and repeated in all databases. Where applicable, MeSH and free text terms were combined to identify relevant studies. The search strategy was developed with the support of a librarian at the LSHTM.

Details on the search strategy are presented in S1 Appendix. Articles in English, French, Spanish, Portuguese, or Italian were considered. A search of the grey literature was not conducted.

### Selection and appraisal of records

Records were selected on predefined inclusion and exclusion criteria guided by the Population, Intervention, Comparison, Outcome and Study design/setting (PICOS) framework (S1 Table) [33]. Inclusion criteria consisted of population (individuals with presumptive pulmonary TB and health care providers involved in TB diagnosis and treatment), findings/outcomes (any relevant to clinical decision-making), and setting (high TB burden countries). We considered relevant to the decision-making any intervention, action, or event that influenced the diagnosis of TB. SRs, meta-analyses, and SRs of qualitative studies (hereinafter referred to as qualitative evidence syntheses) were included. Articles exclusively on drug-resistant TB, non-review articles, and reviews that did not use systematic methods were excluded (S1 Table).

Following removal of duplicates, the title/abstract screening was carried out by a single reviewer (FWB). The full text of selected records was then examined for inclusion in the study, based on the predefined criteria (S1 Table).

## Quality appraisal

Methodological quality, risk of bias and reporting quality of reviews were assessed using the JBI checklist for SRs [28, 30]. No records were excluded on grounds of quality due to lack of consensus on the most appropriate tools and approaches for managing low-quality reviews in umbrella reviews [34] (S2 Table). Where available, GRADE assessments [35, 36] were extracted and reported.

## Overlap assessment

Several approaches have been proposed for overlap management in umbrella reviews [37]. We included all eligible reviews and documented the extent of overlap in primary studies using the Corrected Covered Area (CCA) index [37]. After obtaining the overall CCA, pairwise indexes were calculated (S1 Fig). For reviews with moderate to high pairwise CCA, research aims and reported outcomes were examined. If two reviews had the same aims, findings from the highest quality review were described [37].

## Data extraction

Study characteristics and data of interest for included records were extracted by a single reviewer (FWB) [30, 31, 38]. These included: type of review, title, authors, publication year, number of studies and participants included in the review, aims/objectives/PICO question (or equivalent), search strategy, methodological quality/risk of bias and certainty of evidence assessment. For reviews examining global data, only findings pertinent to high-burden TB settings were extracted. Data extraction also indicated where pooled analyses included non-high TB burden countries.

Primary studies from reviews were not retrieved.

## Data synthesis

Data synthesis used a systematic narrative approach for umbrella reviews [38], which involved thematic content analysis and coding of findings from each review to identify recurring themes associated with factors influencing TB clinical decision-making. Nvivo (version 1.5, 2021, QSR International Pvt. LTD, Australia) was used to iteratively code extracted key data. Themes were developed separately for quantitative and qualitative studies, then combined and presented complementarily [39, 40].

Barriers and facilitators for TB testing or treatment decisions from each review were coded first, and then grouped under common themes associated with decision-making uncertainty, based on the taxonomy developed by Eachempati *et al.* [20]. The taxonomy develops around macro (society and community), meso (group relationships), and micro (individual) levels of uncertainty to emphasize the dynamic interplay between different sources and types of uncertainty at each level, and may be useful to classify challenges in health care decision-making [20].

Recurring themes were further classified based on an adapted version of the WHO conceptual framework representing the TB diagnosis and care continuum [41]. The framework helped to identify four levels (patient, provider, health system, diagnostic test) of factors influencing TB clinical decision-making, including three time-points (patient-provider encounter, diagnosis, treatment initiation) for decision-making. The framework captures both the

determinants (i.e., what causes decisional uncertainty) and broader sources (i.e., what contributes to the variability of decisional outcomes) of uncertainty in the decision-making process.

### Definitions

Presumptive pulmonary TB was defined as clinical/pre-test suspicion or post-test suspicion despite a negative test. Diagnostic delay was defined as the time lag from first access to health system/consultation with provider to diagnosis; treatment delay was defined as the time lag from diagnosis to treatment initiation. Provider/health system delay was used to refer to any diagnostic or treatment delay attributable to provider or health systems factors (to differentiate from causes of delay attributable to patient factors).

## Results

### Review characteristics

Database searches yielded 8542 records. After duplicate removal, 7345 unique records were screened by title/abstract. After full text screening of 110 records, a total of 27 reviews were included (Table 1). The PRISMA flow chart detailing the phases of study selection is presented in Fig 1.

Articles were published between 2008 and 2023. Records included nine meta-analyses, three qualitative evidence syntheses, and 15 mixed-methods narrative synthesis (Table 1). Primary studies included in the reviews spanned from 1970 to 2021 and were mostly observational (Table 1).

Reviews varied in inclusion criteria, outcomes, settings, and population. Based on their primary aim, reviews were classified into four main categories: diagnostic and treatment delays (n = 8); knowledge, attitudes, and practices of TB healthcare providers and end users (n = 5); barriers and facilitators to utilization of TB diagnostic services (n = 10); diagnostic test impact on diagnosis and treatment (n = 4). Most reviews included primary studies with adult populations or did not include sub-group analysis by age. One review focused specifically on children and adolescents [25]. Key population and outcome definitions were generally consistent. Prior to the review, standardized definitions were developed allowing for direct comparison and a narrative synthesis of findings (Table 1).

Most reviews were of fair or good methodological quality. The main areas compromising methodological quality and confidence in findings were publication bias, not using consistent methods to minimize errors in data extraction, and not grading the quality of evidence (S2 Table). Based on the global CCA index, most reviews had very low to no overlap. Seven pairs had high or very high primary source overlap. The citations list from one review [42] was not available, hence not included in CCA calculations (S1 Fig).

### Main findings from thematic content analysis

Through iterative thematic analysis, 15 recurring themes were identified. Applying an integrated multilevel model of uncertainty in health care [20], the themes were classified by type of uncertainty (Table 2). Findings were then classified as barriers or facilitators for testing and treatment decisions (Fig 2).

Synthesis enabled the development of a framework for uncertainty in TB decision-making, presented in Fig 3. Types of uncertainty were grouped in four macro-levels, corresponding to sources of uncertainty in TB clinical decision-making: patient-, provider-, diagnostics-, and health system-related uncertainty. The framework represents the relationship between the four sources of uncertainty and three key moments in clinical decision-making: the clinical

**Table 1. Overview of the studies included in the UR.**

| Review Author, year | Review design | Key findings | Primary studies included in reviews | | | | Critical appraisal [1] (JBI score) |
|---|---|---|---|---|---|---|---|
| | | | Population; Setting | Total included; type (n) | Years | Contributing to UR findings | |
| *Diagnostic and treatment delays[2,3] and associated factors (n = 8)* | | | | | | | |
| *Cai 2015* | Systematic review and meta-analysis | Male sex, older age, lack of education, rural residence were associated with provider delay. | Presumptive TB[4] of all age and sex; any healthcare level, Asia | 45; cross-sectional (n = 43); cohort (n = 2) | 1997–2014 | 30(including 4 non-high burden countries) | Good |
| *Bello 2019* | Systematic review and meta-analysis | Provider delay second most important contributor to delay in TB care. Studies using CXR reported lowest delays compared to sputum culture and microscopy. | Presumptive TB of all age and sex; n/s | 198; n/r | 1983–2014 | n/r | Poor |
| *Getnet 2017* | Systematic review | Sociodemographic/economic risk factors for health system delay: older age, distance from hospital, low income, unemployment. Clinical risk factors: good functional status, no cough, unusual symptoms, normal/fibrotic appearance CXR, fever, smear negativity. Healthcare setting risk factors: private providers, peripheral centers. | Presumptive TB aged 15 and above; any healthcare level, LMIC/LIC | 40; cross-sectional (n = 39), cohort (n = 1) | 2007–2015 | 10 | Fair |
| *Lee 2022* | Systematic review and meta-analysis | The use of mWRD reduced diagnostic delay and time from diagnosis to TB treatment initiation. | Presumptive TB/cases of all age and sex including DR-TB; no setting restrictions | 45; RCT (n = 6);before/after (n = 2); single-arm interventional pilot (n = 1); observational (n = 36) | 2011–2020 | 21 (DS-TB, high burden countries) | Good |
| *Li 2013* | Systematic review and meta-analysis | Risk factors for health system delay: low educational attainment, rural residence, lack of health insurance, low income and inability to afford time off work, traditional healers, low availability of resources, inability to prescribe tests. Shortage of trained/knowledgeable providers important cause of TB provider delay. | Presumptive TB/cases of all age and sex; any healthcare level; China | 29; Cross-sectional (n = 27); Cohort (n = 1); Case-control (n = 1) | 2000–2011 | 29 | Good |
| *Sreeramareddy 2014* | Systematic review | Seeking care from private provider risk factor for diagnostic delay. Distance from health center and seeing multiple providers significantly associated with diagnostic delay. | Presumptive TB/cases[5] of all age and sex; any healthcare level, India | 23; Cross-sectional (n = 21); Cohort (n = 2) | 1998–2013 | 5 | Fair |
| *Storla 2008* | Systematic review | Risk factors for diagnosis delay include: chronic cough in presence of other lung disease, negative smear, rural residence, unqualified provider, female patient, patient alcoholism/substance abuse and patient low educational level. | Presumptive TB/cases of all age and sex; All settings, high and low-income countries | 58; n/r | 1992–2007 | n/r | Poor |
| *Teo 2021* | Systematic review (mixed-methods) | Poor practices and ignorance of TB among health providers at health facilities led to a delay in TB diagnosis; first visit at lower-level facilities positively associated with delay. | Presumptive TB/cases of all age and sex; any healthcare level; high TB-burden countries | 124; Qualitative(n = 36), quantitative/observational (n = 86), mixed-methods (n = 2) | n/r | n/r(meta-analysis); 18(qualitative synthesis) | Good |
| *Knowledge, attitudes, and practices of TB healthcare providers[6] and end users (n = 5)* | | | | | | | |
| *Amare 2023* | Systematic review and meta-analysis | Providing trainings to healthcare workers significantly increased TB detection rates in adult and pediatric populations, and increased the use of diagnostic tools. | TB care providers and volunteers attending at least 3-day training; no setting restriction | 9: Cluster RCT (n = 5), non-RCT (n = 4) | 2005–2016 | 9 | Good |

*(Continued)*

**Table 1.** (Continued)

| Review Author, year | Review design | Key findings | Primary studies included in reviews | | | | Critical appraisal [1] (JBI score) |
|---|---|---|---|---|---|---|---|
| | | | Population; Setting | Total included; type (n) | Years | Contributing to UR findings | |
| *Bell 2011* | Scoping review, narrative synthesis | TB symptom awareness among providers varied by setting. Providers often did not know cough duration criteria that might arouse suspicion of TB and had poor knowledge of guidelines relating to TB diagnostic tests. Knowledge-associated variables included age, sex, location, qualification, employment sector. | TB care providers (both qualified and non-qualified); any healthcare level; high-TB burden countries (WHO 2006) | 34; Cross-sectional (n = 22), participant surveys (n = 8), in-depth interviews (n = 4) | 1998–2009 | 8 | Fair |
| *Engel 2022* | Qualitative evidence synthesis | Providers value the rapidity and accuracy of mWRD, the possibility to use diverse sample types and have confidence in mWRD results for patient management decisions, though overconfidence in mWRD can result in underdiagnosis. Providers can be reluctant to test for TB because of TB-associated stigma and its consequences, fears of acquiring TB themselves, fear of adverse effects of drugs in children. Availability of mWRD. | Users or potential users of NAATS (patients, caregivers, providers, laboratory technicians, TB officers); any settings; high TB and MDR-TB burden countries | 32; Mixed-methods (n = 13), qualitative (n = 19) | 2012–2021 | 18 | Good |
| *Satyanarayana 2015* | Systematic review, narrative synthesis | Among persons with cough of 2–3 weeks' duration, less than two thirds were advised to undergo sputum examination. Adherence to guidelines consistently higher in the public sector. Public sector providers were more likely to know that sputum smear examination is the primary test for TB.(pre-Xpert) | Healthcare providers involved in TB care; Any healthcare level; India | 47; cross-sectional (n = 46); interventional (n = 1) | 2002–2014 | 12 | Fair |
| *Thapa 2021* | Scoping review, narrative synthesis | Appropriate knowledge to deliver health care and IPs knowledge and skills as crucial factors that influences the quality of care. Given their important role in patient care, future research should attempt to measure IPs' knowledge and skills in TB care. | Informal healthcare providers; all healthcare settings; LIC/ LMIC | 13; quasi-experimental (n = 10); clusterRT (n = 1); unclassified (n = 2) | 1980–2019 | 3 | Good |
| *Barriers and facilitators to utilization of TB diagnostic services including testing, diagnosis, and treatment (n = 10)* | | | | | | | |
| *Barnabishvili 2016* | Scoping review, narrative synthesis | Negative attitudes from providers included discriminating and oppressive or even aggressive behavior, triggered by patient gender, age and ethnicity. Barriers for female patients resulted from "several male doctors, describing the meeting with female TB patients as 'difficult'" and their perception that "women present their symptoms in a "less concrete way". | People of all ages and sex; any healthcare level; TB and MDR-TB high-burden countries | 12; Qualitative (n = 11); Secondary data analysis (n = 1) | 2002–2016 | 3 | Fair |
| *Bhatnagar 2019* | Systematic review (mixed-methods), narrative synthesis | Poor provider knowledge resulted in suggesting and treating for incorrect diagnoses. Communication barriers as causes of missed treatment and patient loss to follow-up. | Any presumptive TB or person accessing TB services or end-user aged 15 and above; any healthcare level; India | 39; semi-structured or structured interviews (n = 27), in-depth interviews (n = 7), focus group discussions (n = 3) | 2002–2018 | n/r | Fair |

*(Continued)*

**Table 1.** (Continued)

| Review Author, year | Review design | Key findings | Primary studies included in reviews | | | | Critical appraisal [1] (JBI score) |
|---|---|---|---|---|---|---|---|
| | | | Population; Setting | Total included; type (n) | Years | Contributing to UR findings | |
| *Braham 2018* | Systematic review, narrative synthesis | Less than one every two practitioners knew the importance of sputum microscopy as the main tool needed for TB diagnosis (pre-Xpert data). Use of mWRD was heterogeneous and low, ranging from 0% to 52%. | Any TB practitioner; Any healthcare level; Pakistan | 11; Cross-sectional (n = 11) | 1996–2014 | 6 | Good |
| *Dlangalala 2021* | Scoping review, narrative synthesis | Increasing hesitancy to handle any sputum samples or observe sputum collection in African countries during COVID-19 pandemic. Lack of PPE discouraged staff from attending patients. | n/r; primary healthcare level; worldwide | 21; Primary research (n = 5), Editorials/reports (n = 16) | 2020–2021 | 4 | Poor |
| *Krishnan 2014* | Qualitative evidence synthesis | Women experienced more barriers (including stigma) to accessing TB care than men. Gender-related differences are context-specific. | Presumptive TB aged 15 and above, healthcare providers; any healthcare level; no setting restriction | 28; Qualitative | 1995–2010 | 11 | Good |
| *Oga-Omenka 2021* | Qualitative evidence synthesis (meta-synthesis) | The attitude of healthcare workers created barriers to diagnosis. Unbearable workloads, inadequate training and a lack of laboratory resources were barriers for good diagnostic service provision and delayed access to testing. Health system problems included poor IPC measures, staff shortages, overwhelming workloads, and lengthy triage procedures. Facilitators are patient financial support, quick test turnaround times, appropriate counseling, patient tracking, health worker training, good workflows, adequate staffing, TB services free of charge and private spaces for consultation. | Presumptive TB of all age and sex; Any healthcare level, Nigeria | 10; Qualitative (n = 9), Mixed-methods (n = 1) | 2006–2020 | 10 | Good |
| *Shah 2022* | Scoping review, narrative synthesis | Unavailablity of, or lack of access to, diagnostic tests, and missed diagnosis despite reaching healthcare facilities represented major diagnostic gaps identified in this review. | Any presumptive TB or person accessing TB services or TB service end-user and provider; any healthcare level; worldwide | 61; n/r | 2008–2020 | | Fair |
| *Sullivan 2017* | Systematic review, narrative synthesis | Poor infrastructure was a barrier to treatment. Facilities in rural areas with improved TB diagnostic and treatment capacity could reduce diagnostic and treatment delays. Long test result times delayed treatment for children especially if TB exposure was unknown. | Presumptive TB 0 to 24 years old; all healthcare levels; sub-Saharan Africa | 47(4 pediatric only); n/r | 1994–2015 | n/r | Poor |
| *Yang 2014* | Systematic review | Majority of examined studies found no gender-related difference in provider delays, but there was setting variability underlying the gender-related attitudes of providers. Women generally experienced more barriers than men. | Presumptive TB/cases aged 15 years or older; all settings | 137; Cross-sectional (n = 126); Case-control (n = 1); Cohort (n = 8); RCT (n = 1); cluster-RCT (n = 1) | 1970–2010 | 37 (observational) | Fair |

*(Continued)*

**Table 1.** (Continued)

| Review Author, year | Review design | Key findings | Primary studies included in reviews | | | | Critical appraisal [1] (JBI score) |
|---|---|---|---|---|---|---|---|
| | | | Population; Setting | Total included; type (n) | Years | Contributing to UR findings | |
| *Yasobant 2021* | Systematic review | The review identified several barriers to diagnosis/testing including: overburdened staff, lack of assured specimen transport and tracking, inadequate history taking and misinterpretation of provisional diagnosis, poor attitudes and behavior of providers, poor counselling capacities. | Presumptive TB/cases of all age and sex; India | 28; Quantitative/ observational (n = 19), qualitative (n = 6), mixed-method (n = 3) | 2010–2020 | n/r | Poor |
| *Diagnostic test impact on diagnosis and treatment (n = 4)* | | | | | | | |
| *Agizew 2019* | Systematic review and meta-analysis | Use of Xpert reduced time to treatment and time to treatment compared with smear microscopy. Use of Xpert might be associated with a decrease in empiric treatment. | Presumptive TB/cases of all age and sex; no setting restrictions | 13; RT(n = 9), cohort(n = 4) | 2012–2015 | 6 | Good |
| *Di Tanna 2019* | Systematic review and meta-analysis | Time to diagnosis in the Xpert vs smear group did not differ for 1924 individuals from two studies (very high citation overlap with Agizew et al.) | Presumptive TB/cases of all age and sex; no setting restrictions | 5; RCT (n = 2), CRT (n = 2), SW (n = 1) | n/r | 2 | Good |
| *Haraka 2021* | Systematic review and meta-analysis | Modest or no effect of Xpert on proportion of participants from 5 RCT treated for TB (Moderate confidence). One RCT reported on time-to-treatment initiation (HR 0.76, 95% CI 0.63 to 0.92). | Presumptive TB/cases of all age and sex; no setting restrictions | 12; RCT (n = 3); before/after (n = 4); SW (n = 2); cluster-RCT (n = 3) | 2012–2019 | 6 | Good |
| *Nathavitharana 2021* | Systematic review and meta-analysis | Higher proportion of PLHIV started on tuberculosis treatment when undergoing LAM testing as part of TB cascade vs standard of care. Time to diagnosis was marginally shorter in the LAM group vs standard-of-care. A higher proportion of study participants were able to provide urine specimens instead of sputum. | HIV+ Presumptive TB aged>15years; no setting restrictions | 3 RCT (n = 2); cluster-RCT (n = 1) | 2016–2020 | 3 | Good |

[1] 1 point was assigned for each of the 11 criteria scoring 'yes'. 'Good' indicated reviews that scored 8/11 and above, 'fair' indicated reviews that scored between 5 and 7, 'poor' indicated reviews that scored 4/11 and below.

Prior to the review, standardized definitions for classification and analysis of interventions were developed as follows: [2] Diagnostic delay: time lag from first access to health system/consultation with provider to diagnosis; [3] Treatment delay: time lag from diagnosis to treatment initiation; [4] Presumptive TB: Previously known as TB suspect, any individual not on TB treatment presenting with any sign or symptom suggestive of TB (clinical signs and symptoms used for inclusion in primary studies and reviews may vary) Note: often referred to as TB suspect across reviews and primary studies; [5] TB case: any individual clinically diagnosed or bacteriologically confirmed with TB at the end of the TB cascade or primary study (timepoints may vary across studies and reviews); [6] Provider: any individual delivering health services, and responsible for: formulating diagnoses/diagnostic hypotheses, and/or prescribing diagnostic tests and/or prescribing treatment across various healthcare settings and levels (including informal healthcare providers).

encounter, the formulation of a diagnostic hypothesis, and the treatment initiation. Different types of uncertainty may act synergistically at given time-points (Fig 3).

## Clinical uncertainty

**Clinical presentation.** Half of the reviews mentioned the relationship between clinical presentation and clinicians' suspicion of TB. Meta-analysis data from GeneXpert and urine

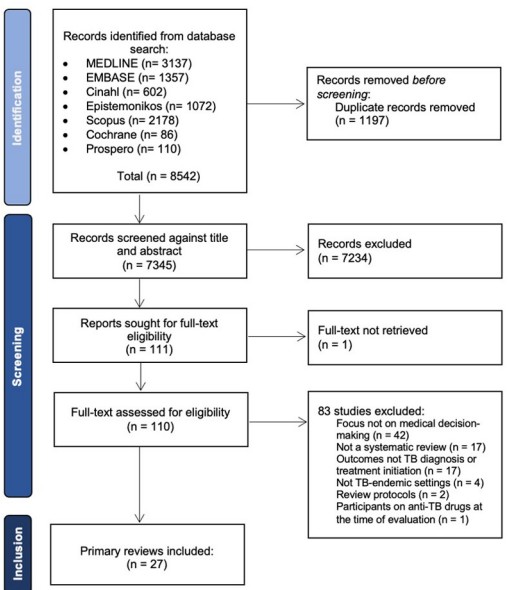

**Fig 1. Study selection flowchart.**

lipoarabinomannan (LAM) diagnostic impact studies suggested a higher likelihood of being treated empirically for sicker patients requiring hospitalization [43, 44]. In the review by Getnet *et al.*, findings from observational studies indicated that the absence of cough and the

**Table 2. Summary of themes and classification by type of uncertainty, applying a multilevel model to TB clinical decision-making.**

| Type of uncertainty | Definition | Source of uncertainty<br>*Level (timepoint)* | Themes |
|---|---|---|---|
| Clinical uncertainty | Uncertainty experienced during clinical encounters, related to diagnostic dilemmas, including those due to variability in clinical presentation, treatment, and prognosis. | *Patient*<br>*(clinical encounter, diagnosis)* | • Clinical Presentation<br>• Socio-demographic attributes<br>• Collection of diagnostic specimens<br>• Risk factors<br>• Side effects |
| Personal uncertainty | Uncertainty generated by personal beliefs, attitudes, fears, experiences, individual risk perceptions and tolerance level. | *Provider*<br>*(clinical encounter, diagnosis, treatment initiation)* | • Attitudes, beliefs, stigma<br>• Fear of infection<br>• Provider preferences of test characteristics |
| Relational uncertainty | Uncertainty arising from the interactions between the different stakeholders in the diagnostic process. | *Provider*<br>*(clinical encounter)* | • Attitudes, beliefs, stigma<br>• Patient-provider communication dynamics |
| Knowledge-exchange related uncertainty | Uncertainty generated by knowledge exchange, such as communication of diagnosis. | *Provider, patient*<br>*(clinical encounter)* | • Patient-provider communication dynamics |
| Epistemic uncertainty | Uncertainty related to quantity and quality of knowledge, including insufficient knowledge because of lack of information. | *Provider, health systems*<br>*(clinical encounter, diagnosis, treatment initiation)* | • Providers' knowledge and qualification<br>• Availability of local policies and guidelines |
| Test uncertainty | Uncertainty due to lack of confidence in test results, or utilization of a test with suboptimal diagnostic accuracy. | *Diagnostics*<br>*(diagnosis, treatment initiation)* | • Utilization and impact of diagnostic tools |
| Health system uncertainty | Uncertainty emerging from the way services/systems are structured, involving complexities of service delivery such as resources constraints. | *Diagnostics, health systems*<br>*(clinical encounter, diagnosis, treatment initiation)* | • Operational setting deficiencies<br>• Availability and timing of test results<br>• Diagnostic tests availability, affordability, and accessibility |

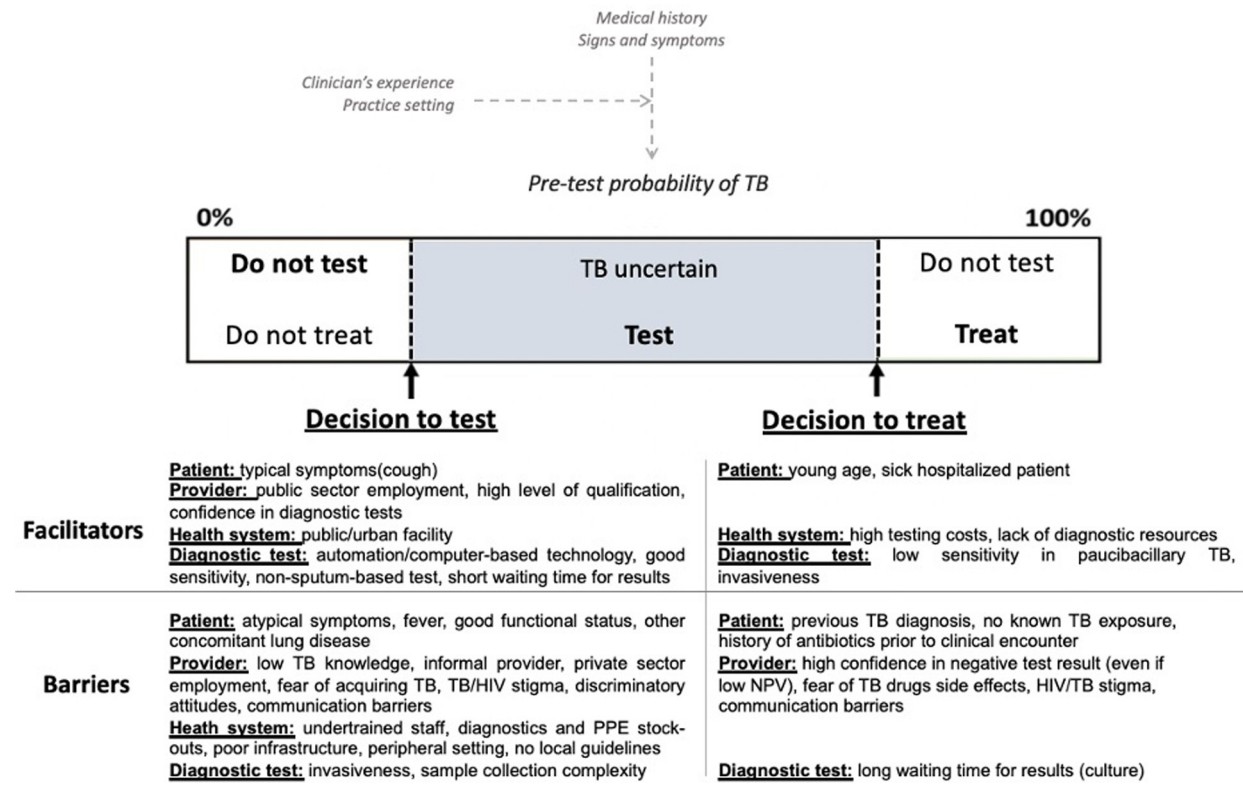

**Fig 2. Facilitators and barriers to TB diagnosis and treatment.**

presence of atypical symptoms, fever, or good clinical conditions were associated with provider diagnostic delay [45]. Similarly, patients presenting with chronic cough and other concomitant lung disease including COVID-19 were reported to experience delays [24, 46].

**Socio-demographic characteristics.** Three reviews identified several indicators of patient socio-economic status, including poor literacy, low income or unemployment, lack of health insurance, and rural residence, as factors associated with diagnostic delay [45, 47, 48]. The reviews by Yang *et al.* and Krishnan *et al.*, which focused on gender-related differences in access to TB services and had moderate overlap, reported inconsistent evidence of a positive relationship between female sex and provider delay in TB diagnosis [47, 49]. Yang *et al.* additionally reported differences by setting. For example, providers from Thailand and Vietnam were more likely to adhere to diagnostic guidelines with male patients, whereas providers from India offered testing with similar frequency to both women and men [49]. The meta-analysis by Getnet *et al.* showed no evidence of a difference in the proportion of male versus female patients diagnosed with TB at the 30-day mark (pooled odds ratio (OR) = 1.08, 95% CI 0.95–1.23) [45]. Similarly, Li *et al.* reported no evidence of an association between female sex for patients and diagnostic delay in China (pooled OR = 1, 95%CI 0.83–1.22) [48].

**TB-related risk factors.** Treatment was often delayed in patients with a previous diagnosis of TB [50] and in patients who reported antibiotic usage prior to the clinical encounter [51]. Among paediatric patients, providers were less likely to start empiric treatment in cases with unknown TB exposure [25].

**Collection of diagnostic specimens.** Reviews reported that the inability of patients to produce sputum influenced decisions to initiate testing and treatment [52]. In Nathavitharana *et al.*, the proportion of adults able to provide a sputum sample ranged between 57% and 97%

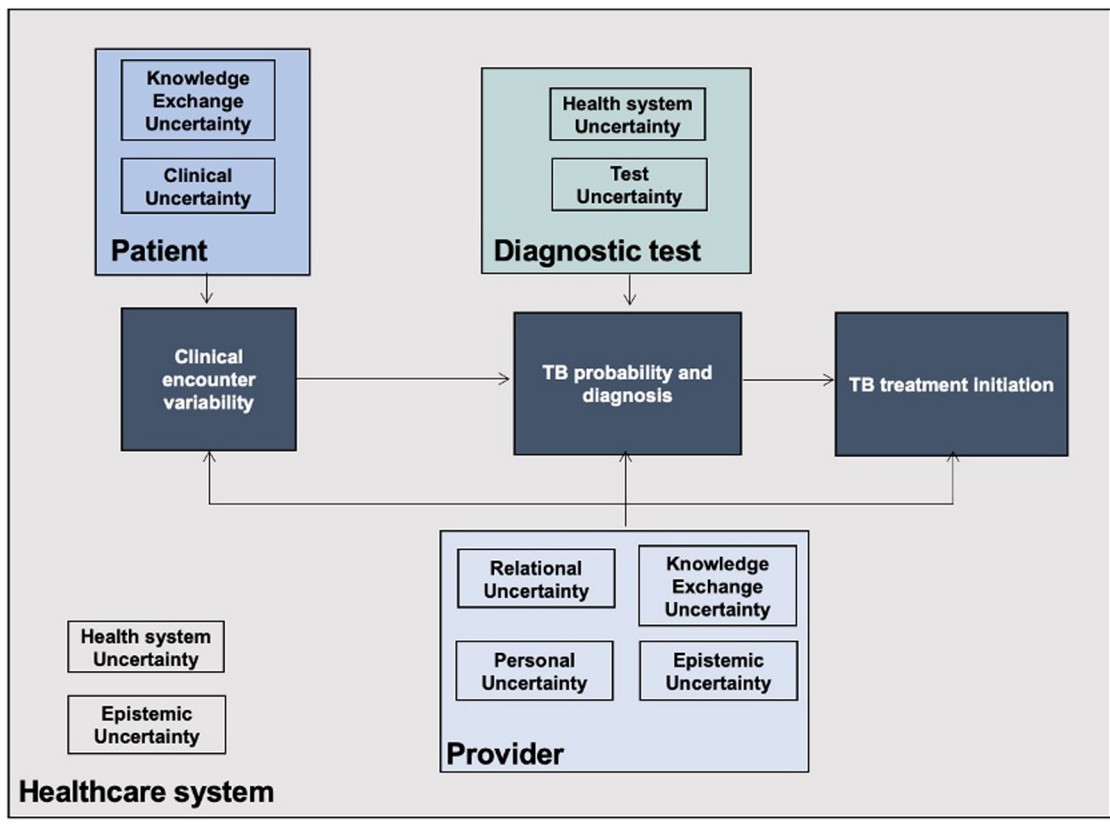

**Fig 3. Conceptual framework for uncertainty in TB clinical decision-making.**

in people living with HIV (PLHIV), depending on setting and severity/type of symptoms. In contrast, urine collection (for the LAM assay) was achieved in 99% of PLHIV aged 15 and above across three RCTs [43]. Challenges with specimen collection influenced decisions to withhold microbiological testing and either initiate empiric treatment or to exclude TB solely based on the clinical interview or radiological examination findings [52, 53]. Engel *et al.*, found with high confidence in evidence that providers highly valued the possibility of using alternative samples for testing such as urine or stool, particularly for paucibacillary cases and paediatric TB [52].

**Side effects.** Reviews also reported provider decisions to withhold or delay treatment initiation because of fear of TB drug side effects in children [25, 52].

### Personal uncertainty

**Provider attitudes, beliefs, and stigma.** Multiple reviews found that provider behaviour and discriminatory attitudes can impact TB diagnosis and treatment initiation [25, 39, 49, 52–57]. In a qualitative evidence synthesis, Barnabishvili *et al.* reported that providers were less rigorous when interviewing older patients or foreigners during the clinical encounter [39]. Provider discrimination towards female patients, resulting in tests underutilization and delays, emerged from narrative syntheses [39, 45, 48, 51]. Provider TB/HIV coinfection-related stigma was reported in three reviews as a factor delaying diagnosis or treatment initiation [52, 53, 55], including one review with high confidence in evidence [52].

**Fear of infection.**  Two reviews based on qualitative data, including one review with high confidence in evidence [52], reported that providers were generally aware of the aerosol bio-hazard and hesitant to test for TB because of fear of acquiring the disease [51, 52]. Fear of infection from respiratory specimen collection, particularly gastric aspiration, resulted in underutilization of diagnostic tools [52] or collection of poor quality respiratory specimens [51, 52]. In the context of the SARS-CoV-2 pandemic, some providers refused to collect respiratory specimens among presumptive TB patients presenting with COVID-19 symptoms [46].

**Test characteristics and provider preference.**  Diagnostic accuracy, automation, and computer-based tests were highly valued by providers based on moderate confidence in evidence [52]. Among paediatric patients, difficulties in collecting respiratory specimens (e.g., induced sputum or gastric aspirate), invasiveness of the procedure, and the lack of adequately trained staff were reported as barriers to test utilization [52, 58].

## Relational and knowledge exchange uncertainty

**Patient-provider communication dynamics.**  Some reviews reported that provider miscommunication with patients was a potential cause of missed diagnoses [39, 42, 51–54]. The difficulty in communicating with the patient was often reported as the consequence of TB-related stigma, but it also arose from the use of metaphors for clinical explanations, resulting in patients not understanding diagnostic and therapeutic plans, and losses to follow up [53, 54]. One review reported that male providers disclosed difficulties communicating with, and understanding health concerns from, female patients during consultation [39].

## Epistemic uncertainty

**Provider knowledge and qualification.**  Qualitative findings from twelve reviews suggested that suboptimal TB knowledge impacted providers' ability to prescribe diagnostic tests or caused providers to delay TB diagnosis and miss treatment opportunities [24, 25, 48, 50, 52, 54, 56, 57, 59–62]. In a review on practices and knowledge of Indian providers, Satyanarayana et al. reported that the proportion of providers that suspected TB in the presence of a persistent cough of more than 2–3 weeks duration ranged from 21% to 81%, and less than 60% of patients with persistent cough were advised to undergo sputum examination [59].

A review by Teo *et al.* reported that poor clinical standards and low levels of knowledge of TB among providers led to delays in TB diagnosis in 12 qualitative studies, with high confidence in evidence [50]. Braham et al. reported one primary study where less than 50% of providers were aware of the principal diagnostic tools needed for TB diagnosis [61]. Poor TB knowledge and clinical skills resulted in deferral of bacteriological testing and preference for smear microscopy over mWRD, according to the narrative review by Shah *et al.*[57]. Additionally, the same review reported that providers' unawareness and non-adherence to diagnostic algorithms was a reason for missed diagnoses [57].

Health care workers with particularly low levels of knowledge included informal providers [62], public providers working at the primary level, private practitioners with limited awareness of TB, and traditional healers [23, 24]. One review found that recognition of TB symptoms was associated with providers' level of qualification and public sector employment [42]. In contrast, age, sex, years of practice, experience, and level of qualification were not associated with identification of TB symptoms [42, 54].

The meta-analysis by Amare *et al.* of nine intervention trials demonstrated that training interventions improve the ability of providers to diagnose TB, significantly increasing the number of bacteriologically confirmed cases [60].

**Availability of policies and guidelines.** Lack of clear and updated guidelines and poor dissemination at primary healthcare levels and among private providers led to poor referral to GeneXpert testing, or inconsistency in the types of samples used [52, 57, 62]. The review by Shah et al. reported guidelines and policies variability in the private sector as one cause of missed diagnoses [57].

## Test uncertainty

**Utilization and impact of diagnostic tools.** Engel *et al.* found that in settings where low-complexity mWRDs were easily accessible, providers reported a high level of trust in the test result [52]. The meta-analysis by Lee *et al* reported that availability of mWRDs reduced diagnostic and treatment delays [63]. Three meta-analyses examined GeneXpert diagnostic impact [44, 58, 64], with outcomes reported only from the most recent review [44]. The use of GeneXpert (versus smear microscopy) had no effect on the proportion of participants treated for TB (risk ratio 1.10, 95% CI 0.98–1.23; GRADE: moderate confidence) [44]. This could reflect decisions to treat some patients empirically regardless of test results. The lower sensitivity of GeneXpert in paucibacillary forms of the disease, such as paediatric TB, was recognized as a limitation that would justify empiric treatment initiation [58].

## Health-system uncertainty

**Operational setting deficiencies.** Twelve reviews reported on challenges at the health systems level. Inadequate staff trainings, lack of diagnostic resources, lack of personal protective equipment and infection prevention control measures, and absence of private rooms for clinical assessment were mentioned as potential contributors to missed diagnosis and treatment opportunities [25, 42, 46, 51, 55–57]. Private and rural clinics not offering TB services were associated with diagnostic delays compared with public, urban facilities, where providers had better access to tests and infrastructure [39, 50, 53, 61].

**Availability and timing of test results.** Sullivan *et al.* reported missed treatment opportunities in children due to long waiting times for culture results [25]. Reviews found that rapid test turnaround time was important to accelerate therapeutic decisions [25, 52, 56], and that offering same-day test and treat would reduce gaps in missed treatment according to providers [52].

**Diagnostic test availability, accessibility, and affordability.** Reviews reported that the limited availability of resources for microbiological diagnosis (e.g., due to stock-outs, power cuts, and unreliable supply chains) was associated with GeneXpert underutilization and diagnostic delays [48, 51, 52, 57]. Engel *et al.* reported on providers' perspectives regarding the impact of diagnostic accessibility and affordability on test and treatment decisions. Frequent stock-outs were reported to potentially hinder providers' faith in the adoption of new diagnostics and hamper their reliance on prescribing diagnostic tests in the future [52]. Further, some providers disclosed a preference for initiating treatment if patients incurred excessive costs for testing, regardless of test availability [52].

## Discussion

This umbrella review showed the complexity of multi-level factors that contribute to uncertainty in TB clinical decision-making, often resulting in under-utilization of diagnostic resources, misdiagnoses, empirical treatment or missed treatment opportunities, and diagnostic and treatment delays. The results of this study reinforce the concept that clinical decision-making is highly dependent on individual and interpersonal factors (provider, patient), but also closely linked to the operational context and the usability of diagnostic resources. These findings are important to inform the development of successful diagnostic aids and programs

implementation strategies, and to improve TB practices in high-burden, resource-limited settings.

An important output from this study was the consolidation of a framework to present multilevel factors associated with uncertainty in TB decision-making. We found that several factors related to the local context and often beyond providers' control were responsible for the discrepancy between TB testing and treatment decisions and scientific guidelines' recommendations. Most of the existing literature on TB diagnostics includes diagnostic accuracy studies or randomized controlled trials that do not examine the challenges of clinical decision-making and the impact of health systems factors on diagnostic interventions. Rapid molecular diagnostics such as GeneXpert have had a great influence on TB care but there are ongoing concerns about underutilization and sustainability that need to be addressed [6]. Unfortunately, diagnostic tests, despite being cheap, fast, and accurate, are not always used as recommended–or not used at all–in high-burden settings, and it is crucial to increase our understanding of the underlying reasons [8, 65].

Reviews reported consistent evidence for patient characteristics, symptom variability and severity as primary sources of clinical uncertainty in TB decision-making [24, 47, 54, 59, 61]. When confronted with hospitalized patients, patients with advanced HIV disease, or paediatric patients, providers seemed more inclined to treat empirically, regardless of the availability of diagnostic aids, possibly also because of the complexity of obtaining clinical specimens from people in these categories [25, 43]. Additionally, history of previous TB diagnosis was associated with retreatment delays [50], potentially due to lack of confidence in diagnosis, or fear of drug side effects with injectables [66]. Further research is needed to uncover provider-related factors associated with retreatment decision-making, as rapid tests for second-line drug resistance testing and all-oral regimens become available [67, 68].

Providers' limited knowledge of TB symptoms and approaches for clinical and diagnostic management, and insufficient familiarity with guidelines, were reported consistently as key contributors to delay in test and treatment decisions [24, 25, 48, 50, 52, 54, 56, 57, 59–62]. Epistemic uncertainty affected several aspects of the decision-making, including estimating pre-test disease probabilities, deciding to use a diagnostic test, selecting appropriate specimens based on age and disease localization, collecting good quality samples, and interpreting test results [50, 59, 61]. Conversely, the availability of highly qualified physicians, public sector facilities, and ease of access to mWRD had a positive influence on testing decisions [42, 52]. Notably, training interventions significantly improved case detection and test uptake by providers [60].

The central role of the provider in the decision-making process was also supported by extensive evidence on how interpersonal attitudes, beliefs, stigma, fear of infection, and test preferences affected test utilization and treatment decisions [25, 39, 49, 52–57]. Personal sources of uncertainty, including fear of acquiring TB through respiratory sampling, were commonly reported barriers for underutilization of diagnostics [51, 52]. As seen with other respiratory infectious diseases, fear of infection was mostly associated with poor knowledge of biohazard mitigation strategies, ambiguous guidelines, and lack of resources [69]. These findings support the importance of enhancing comprehensive national training and educational programs for providers at all levels of care, and engaging the private sector [61, 70]. Similarly, the fear of acquiring TB could be, at least partially, addressed through continuous training, and implementation of infection prevention control measures [71].

The high variability of provider-patient interactions during the clinical encounter was often reported as a source of relational uncertainty affecting the outcomes of the clinical decision-making process [39, 42, 53]. Provider personal biases could result in the inability or unwillingness to collect all necessary clinical information, diagnostic test under-utilization, misdiagnosis

and diagnostic delays, especially when confronted with female patients [48, 49, 51, 61]. Although findings from meta-analyses did not confirm the association between female sex and diagnostic delays, moderate-quality qualitative sources reported the impact of gender on clinical decision-making [48, 49, 51]. Gender-related disparities in TB are well-known, especially with regards to health seeking behaviours and retention in care [51]. While TB incidence is greater in men [72], women generally face additional barriers related to care access, stigma and psychosocial consequences of the diagnosis [51]. The findings from this study confirm the importance of a gender-based approach to TB as advocated by WHO [73]. At the same time, quantitative and qualitative studies across settings and countries with different gender norms are needed to gain further insight on gaps in the TB diagnostic cascade, gender inequalities and discrimination, to inform TB interventions that have the capacity to overcome gender barriers [74].

Providers had high confidence in rapid diagnostic tests, but the confidence in mWRDs, namely GeneXpert, appeared to be generated by trust in a computer-based test, rather than from understanding of the technology and knowledge about its diagnostic accuracy [43, 52]. It should be noted that, paradoxically, a blind use of diagnostics could represent a double-edged sword, if overconfidence in results became a substitute for clinical reasoning [75]. The burden of misdiagnosis was also supported by findings from a large autopsy study, demonstrating a high prevalence of TB among children and PLHIV that were missed at clinical diagnosis [76]. Evaluating the impact of testing on clinical decisions and empiric treatment [77, 78] will be important as missing false negative patients contributes to TB morbidity and mortality, particularly among people who cannot expectorate or who have paucibacillary disease such as young children, where currently available assays have lower sensitivities [79–81]

Health system uncertainty emerged as an important driver of variability in TB decision-making. The unavailability or inaccessibility of diagnostic resources contributed to uncertainty in the decisional process and outcomes [25, 52, 55, 56].

When diagnostic tests were available, several contextual factors, such as poor infrastructure and lack of administrative resources (infection prevention control policies, insufficient trainings), represented barriers to test adoption, shifting the decisional bar towards empiric treatment initiation, particularly in children or very sick patients, or leading to missed treatment opportunities [25, 52]. The absence of locally tailored guidelines was reported to contribute to epistemic uncertainty and variability in clinical management [52, 62]. These findings confirm that resource allocation strategies, as well as trainings and guidelines, need to be more inclusive of the lower tiers of the health system [82].

This study also found that providers highly valued the possibility to use non-sputum samples for testing, such as urine or stool [52], highlighting the need for a rapid addition of sputum-free diagnostics, particularly for paucibacillary cases and paediatric TB [83].

In recent years, there has been unprecedented development of novel TB diagnostic technologies. As new products come to market, policy makers must decide which available tools to implement. Findings from this review support the idea that such decisions should not exclusively account for diagnostic assay characteristics (e.g., accuracy), but also consider acceptability and feasibility of tests within the health care infrastructure. As suggested by meta-analyses reporting inconclusive findings regarding the impact of GeneXpert on treatment initiation decisions [44, 64], it is key to understand the real-world impact of diagnostics through robust operational research at the point-of-care.

Additionally, the increasing utilization of multiple tests or different specimens in parallel, may exacerbate the challenges of results interpretation, particularly in children [84]. Understanding how clinicians manage conflicting results will be important to inform clinical algorithms.

Recently, significant progress has been made in the development and validation of clinical prediction models and algorithms to help standardize the decision-making process, particularly in contexts not yet reached by new diagnostic tools [85]. However, such tools rely on the assumption that a clinical consultation is a standardized event where relevant clinical variables or risk factors would always be disclosed and inform disease probability. Nonetheless, as suggested by the findings of this review, a clinical encounters is an event influenced by multiple uncertainties [39, 53, 55, 56]. Hence, it will be important to collect data on real-life performance of such prediction models and algorithms, and to consider setting-specific adjustments and the integration of variables beyond patient clinical and risk factors. At the same time, the complex roots of uncertainty call for integrated efforts by policy makers, researchers, and programs to combine diagnostics research and implementation with staff trainings, guidelines implementation and uptake, infrastructure development, transversal health education to combat stigma and discrimination, and investments at the most peripheral levels of health care systems globally.

## Strengths and limitations

To the best of our knowledge, this is the first study to conceptualize and summarize sources and types of uncertainty in TB decision-making. The umbrella review approach allowed us to triangulate findings from varied study designs and outcomes while preserving high methodological standards. The review was conducted in a systematic manner in accordance with standardized guidance. Nonetheless, some limitations must be mentioned. First, a limitation of the umbrella review approach is our inability to conduct a detailed assessment of primary studies. Consequently, the study relied on the methods and quality of included SRs, many of which were of moderate quality. Most reviews used a narrative synthesis approach, and only a few meta-analyses and one qualitative evidence synthesis reported on the quality of the evidence. Second, it was not possible to perform a meta-analysis of quantitative review findings due to the heterogeneous inclusion criteria and outcome definitions. Third, it is possible that some relevant sources were missed, as grey literature was not included. Finally, the assessment of each record was performed by a single reviewer only, which may yield a lower sensitivity.

## Conclusion

This study summarized the complex network of factors associated with decisional and outcome uncertainty in medical decision-making in TB through a synthesis and thematic analysis of the systematic review literature. Different sources of uncertainty were found to influence provider choices around testing and treatment initiation, often resulting in diagnostic and treatment delays or missed diagnoses and treatment opportunities. Further, the application of a multi-level framework to classify uncertainty revealed the extent to which findings pertaining to different sources and types of uncertainty were intertwined. Gaps in TB diagnosis and treatment suggest the need to integrate evidence from studies that consider variations in healthcare systems and end-users' attitudes, preferences, and experiences with interventions introducing new diagnostic tools. Such considerations are important to improve TB diagnosis and treatment and quality of patient care and to allow impactful introduction of novel diagnostic aids in clinical practice worldwide.

The figure summarizes multi-level (patient, provider, health systems, diagnostic tests) factors associated with TB clinical decision-making, identified through thematic content analysis of the SRs. The factors were classified as barriers or facilitators for testing or treatment decisions, and represented using the threshold model [10]. Several facilitators positively influenced providers' decisions to test (lower testing threshold), including the presence of typical

symptoms and patient history, providers' personal attributes and experiences, workplace (public/urban facility), and available test characteristics. Barriers to testing were the presence of confounding/atypical symptoms, inadequate TB knowledge and staff training, fear of infection, lack of resources, and challenges of respiratory specimen collection. Empiric treatment decisions (treatment threshold) were facilitated by the presence of factors generally associated with an increased risk of severe disease or negative outcomes (young age, severe symptoms), unavailability or inaccessibility (e.g., because of costs) to diagnostic tests, and lack of confidence in tests with low sensitivity. Providers were inclined to withhold treatment decisions if facing with certain elements of patient history (e.g., unknown TB exposure), waiting for test results, and in the presence of negative test results (without considering the possibility of a low negative predictive value).

During the clinical encounter, the provider assesses the patient's clinical variables (clinical uncertainty) to determine the disease probability and evaluate therapeutic benefit-harm trade-offs. Disease probability estimates depend on the provider's knowledge and experience (epistemic uncertainty). Provider's ability to conduct an informative, high-quality clinical assessment is also influenced by patient-provider relation and communication strategies (relational and knowledge exchange uncertainty) as well as by provider's attitudes and beliefs (personal uncertainty). When a decision is made to test, the probability of disease is adjusted based on diagnostic test results (post-test probability). However, a negative test result may be insufficient to withhold therapy, considering the low sensitivity of currently available diagnostic tests and the potential benefit of empiric treatment (test uncertainty). Additionally, the provider may decide not to proceed with invasive specimen collection and testing because of individual risk assessments such as fear of infection (personal uncertainty). Thus, the characteristics of diagnostic tests can impact decision-making. Clinical decisions are further limited by healthcare setting constraints such as lack of skilled staff, poor infrastructure, and scarcity of diagnostic tools (health system uncertainty), and absence of local guidelines (epistemic uncertainty).

## Supporting information

**S1 Appendix. Search strategy (July 2023).** The search strategy was refined and tested in the MEDLINE database, and then adapted to the other databases. To restrict the search to capture SRs, while simultaneously minimizing the capture of non-SR publications, search terms and MeSH specific to SR study designs (e.g. 'SR', 'qualitative evidence synthesis') were included. A search of grey literature was not conducted. Details on the search strategy across all databases are presented in S1 Appendix.
(DOCX)

**S1 Table. Eligibility criteria- PICOS framework.** SRs were selected on predefined inclusion and exclusion criteria guided by the Population, Intervention, Comparison, Outcome and Study design/setting (PICOS) framework. References were pre-emptively de-duplicated in Endnote. The selected references were imported in Covidence (https://www.covidence.org/home) and re-screened for duplicates. Full texts of all potentially eligible reviews were obtained. The full text of selected papers was then examined for inclusion in the UR, based on the predefined criteria. The reason for the exclusion of each article was documented in the software for transparency and auditing purposes.
(DOCX)

**S2 Table. Critical appraisal of methodological quality of included reviews.** JBI Critical Appraisal Checklist for Systematic Reviews and Research Syntheses.
(DOCX)

**S1 Fig. Heatmap showing pairwise calculation of the CCA.** CCA was interpreted in banded thresholds: values below 5% indicated slight citations overlap, between 6–10% indicated moderate overlap, between 11–15% indicated high overlap and values above 15% indicated very high overlap.
(DOCX)

**S1 Checklist. PRISMA 2020 checklist.**
(DOCX)

**S2 Checklist. Research checklist, PRISMA 2020 checklist.**
(DOCX)

## Author Contributions

**Conceptualization:** Francesca Wanda Basile, Sedona Sweeney, Ted Cohen, Nicolas A. Menzies, Anna Vassall, Pitchaya Indravudh.

**Data curation:** Francesca Wanda Basile.

**Formal analysis:** Francesca Wanda Basile.

**Funding acquisition:** Anna Vassall.

**Methodology:** Francesca Wanda Basile, Sedona Sweeney, Maninder Pal Singh, Anna Vassall, Pitchaya Indravudh.

**Project administration:** Sedona Sweeney, Pitchaya Indravudh.

**Supervision:** Else Margreet Bijker, Pitchaya Indravudh.

**Writing – original draft:** Francesca Wanda Basile.

**Writing – review & editing:** Francesca Wanda Basile, Sedona Sweeney, Maninder Pal Singh, Else Margreet Bijker, Ted Cohen, Nicolas A. Menzies, Anna Vassall, Pitchaya Indravudh.

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
