## [Decision Letter · Decision Letter 0]

23 Feb 2024

PGPH-D-23-02516

Uncertainty in tuberculosis clinical decision-making: an umbrella review with systematic methods and thematic analysis

Dear Dr. Basile,

Thank you for submitting your manuscript to PLOS Global Public Health. After careful consideration, we feel that it has merit but does not fully meet PLOS Global Public Health’s publication criteria as it currently stands. Therefore, we invite you to submit a revised version of the manuscript that addresses the points raised during the review process.

We look forward to receiving your revised manuscript.

Kind regards,

Sok King Ong

Academic Editor

Journal Requirements:

Additional Editor Comments (if provided):

Reviewers' comments:

Reviewer's Responses to Questions

**Comments to the Author**

1. Does this manuscript meet PLOS Global Public Health’s publication criteria? Is the manuscript technically sound, and do the data support the conclusions? The manuscript must describe methodologically and ethically rigorous research with conclusions that are appropriately drawn based on the data presented.

Reviewer #1: Partly

Reviewer #2: Yes

2. Has the statistical analysis been performed appropriately and rigorously?

Reviewer #1: N/A

Reviewer #2: I don't know

3. Have the authors made all data underlying the findings in their manuscript fully available (please refer to the Data Availability Statement at the start of the manuscript PDF file)?

Reviewer #1: No

Reviewer #2: Yes

4. Is the manuscript presented in an intelligible fashion and written in standard English?

Reviewer #1: Yes

Reviewer #2: Yes

5. Review Comments to the Author

Reviewer #1: Major comments

The availability and access to diagnostic testing and the decision to test when available are two distinct concepts although there is overlap I feel at times these two concepts are being discussed interchangeably in the manuscript, it seems the objective and review focus is more around decision to test not availability of testing therefore suggest an attempt is made to be clear about these two concepts.

Minor comments

Lines 66 & 67: best practices are not implemented (e.g., clinicians choosing quick symptom relief with low-cost pharmaceuticals over diagnostic certainty.(7)

-formatting issue here where the parantheses not closed, but also there are many reasons why best practices are not implemented therefore, perhaps this hypothesis could be better phrased as it reads more as fact, esp. given that the focus of the review is to understand the facilitators and barriers…

-perhaps challenges around conflicting test results could also be mentioned, as low sensitivity and specificity in many TB tests can lead to inconsistent results when parallel testing approaches are applied for example with LAM, or DST.

-Not clear how the initial scoping review was conducted.

-Why was diagnostics/testing not included as a broad search strategy or domain?

-were studies conducted in non TB endemic settings excluded? How was TB endemic settings defined? For instance in some high income countries with low TB burden, there are key populations where Tb remains endemic would these have been included/excluded?

-please clarify why only TB patients and providers were inclusion criteria I would have thought given the focus on decision to test that you would want to include individuals with signs and symptoms of TB, but not restricting to only TB confirmed patients. Does this mean that providers of care for non TB patients were excluded? I assume not, perhaps these inclusion exclusion criteria could be further clarified in terms of how they were applied to the studies found in the initial search.

-Why was DRTB excluded? I would think this a critical area to understand facilitators and barriers around decision not to test.

-Was all of the extraction and inclusion criteria assessed by only one reviewer? Is this not a deviation from the PRISMA guidelines?

Reviewer #2: This is a useful summary of the literature in a complex area where the effect of a diagnostic may be influenced by many (unexpected) factors. The text generally reads well and the discussion appears reasonable.

There is one aspect that although mentioned I feel does not receive the attention it may merit I would like the authors to consider:

In the conclusion of the abstract it is said (line 41) “This could result in delayed or missed diagnosis and treatment opportunities.”. This is correct but the lack of efficient diagnostics can also result in inappropriate treatment i.e. over treatment of people who do not have tuberculosis. I feel this is a critical point, that better diagnostics in theory might not increase the number of people diagnosed or treated but none the less increase the number of people correctly treated. This effect is mentioned in table 1 Agizew 2019 “Use of Xpert might be associated with a decrease in empiric treatment.” and also on line 386-87 “This could reflect decisions to treat some patients empirically regardless of test results.”. Also a somewhat similar effect to that seen with improved cancer screening might be expected with more accurate diagnostics; where improvements in cancer screening may paradoxically result in worse outcomes (for the group treated not the population) as treatment is targeted at the correct (ill) population and overdiagnosis is reduced. I feel the two papers mentioned below may be relevant for this discussion:

Noé, A., Ribeiro, R. M., Anselmo, R., Maixenchs, M., Sitole, L., Munguambe, K., ... & García-Basteiro, A. L. (2017). Knowledge, attitudes and practices regarding tuberculosis care among health workers in Southern Mozambique. BMC pulmonary medicine, 17, 1-7.

Garcia-Basteiro, A. L., Hurtado, J. C., Castillo, P., Fernandes, F., Navarro, M., Lovane, L., ... & Martínez, M. J. (2019). Unmasking the hidden tuberculosis mortality burden in a large post mortem study in Maputo Central Hospital, Mozambique. European Respiratory Journal, 54(3).

6. PLOS authors have the option to publish the peer review history of their article (what does this mean?). If published, this will include your full peer review and any attached files.

**Do you want your identity to be public for this peer review?** For information about this choice, including consent withdrawal, please see our Privacy Policy.

Reviewer #1: No

Reviewer #2: **Yes: **Richard M Anthony

---

## [Decision Letter · Decision Letter 1]

18 Jun 2024

Uncertainty in tuberculosis clinical decision-making: an umbrella review with systematic methods and thematic analysis

PGPH-D-23-02516R1

Dear Dr. Basile,

We are pleased to inform you that your manuscript 'Uncertainty in tuberculosis clinical decision-making: an umbrella review with systematic methods and thematic analysis' has been provisionally accepted for publication in PLOS Global Public Health.

Best regards,

Sok King Ong

Academic Editor

Reviewer Comments (if any, and for reference):

Reviewer's Responses to Questions

**Comments to the Author**

1. If the authors have adequately addressed your comments raised in a previous round of review and you feel that this manuscript is now acceptable for publication, you may indicate that here to bypass the “Comments to the Author” section, enter your conflict of interest statement in the “Confidential to Editor” section, and submit your "Accept" recommendation.

Reviewer #1: All comments have been addressed

Reviewer #2: All comments have been addressed

2. Does this manuscript meet PLOS Global Public Health’s publication criteria? Is the manuscript technically sound, and do the data support the conclusions? The manuscript must describe methodologically and ethically rigorous research with conclusions that are appropriately drawn based on the data presented.

Reviewer #1: Yes

Reviewer #2: Yes

3. Has the statistical analysis been performed appropriately and rigorously?

Reviewer #1: Yes

Reviewer #2: I don't know

4. Have the authors made all data underlying the findings in their manuscript fully available (please refer to the Data Availability Statement at the start of the manuscript PDF file)?

Reviewer #1: Yes

Reviewer #2: Yes

5. Is the manuscript presented in an intelligible fashion and written in standard English?

Reviewer #1: Yes

Reviewer #2: Yes

6. Review Comments to the Author

Reviewer #1: Thank you for considering and addressing comments in this revised version.

Reviewer #2: The authors have revised their manuscript as requested and extended the discussion of the effects on diagnostics on inappropriate treatment (over and under treatment).

7. PLOS authors have the option to publish the peer review history of their article (what does this mean?). If published, this will include your full peer review and any attached files.

**Do you want your identity to be public for this peer review?** For information about this choice, including consent withdrawal, please see our Privacy Policy.

Reviewer #1: No

Reviewer #2: **Yes: **Richard Anthony
